# Characterization of Thymoquinone-Sulfobutylether-β-Cyclodextrin Inclusion Complex for Anticancer Applications

**DOI:** 10.3390/molecules28104096

**Published:** 2023-05-15

**Authors:** Eltayeb E. M. Eid, Amer A. Almaiman, Salah Abdalrazak Alshehade, Wardah Alsalemi, Sareh Kamran, FakhrEldin O. Suliman, Mohammed Abdullah Alshawsh

**Affiliations:** 1Department of Pharmaceutical Chemistry and Pharmacognosy, Unaizah College of Pharmacy, Qassim University, Unaizah 51911, Saudi Arabia; 2Unit of Scientific Research, Applied College, Qassim University, Unaizah 51911, Saudi Arabia; 3Department of Pharmacology, Faculty of Medicine, Universiti Malaya, Kuala Lumpur 50603, Malaysia; 4Department of Chemistry, College of Science, Sultan Qaboos University, P.O. Box 36, Muscat 123, Oman

**Keywords:** thymoquinone, sulfobutylether-β-cyclodextrin complex, formulation, SBE-β-CD, anti-cancer, characterization

## Abstract

Thymoquinone (TQ) is a quinone derived from the black seed *Nigella sativa* and has been extensively studied in pharmaceutical and nutraceutical research due to its therapeutic potential and pharmacological properties. Although the chemopreventive and potential anticancer effects of TQ have been reported, its limited solubility and poor delivery remain the major limitations. In this study, we aimed to characterize the inclusion complexes of TQ with Sulfobutylether-β-cyclodextrin (SBE-β-CD) at four different temperatures (293–318 K). Additionally, we compared the antiproliferative activity of TQ alone to TQ complexed with SBE-β-CD on six different cancer cell lines, including colon, breast, and liver cancer cells (HCT-116, HT-29, MDA-MB-231, MCF-7, SK-BR-3, and HepG2), using an MTT assay. We calculated the thermodynamic parameters (ΔH, ΔS, and ΔG) using the van’t Holf equation. The inclusion complexes were characterized by X-ray diffraction (XRD), Fourier transforms infrared (FT-IR), and molecular dynamics using the PM6 model. Our findings revealed that the solubility of TQ was improved by ≥60 folds, allowing TQ to penetrate completely into the cavity of SBE-β-CD. The IC_50_ values of TQ/SBE-β-CD ranged from 0.1 ± 0.01 µg/mL against SK-BR-3 human breast cancer cells to 1.2 ± 0.16 µg/mL against HCT-116 human colorectal cancer cells, depending on the cell line. In comparison, the IC_50_ values of TQ alone ranged from 0.2 ± 0.01 µg/mL to 4.7 ± 0.21 µg/mL. Overall, our results suggest that SBE-β-CD can enhance the anticancer effect of TQ by increasing its solubility and bioavailability and cellular uptake. However, further studies are necessary to fully understand the underlying mechanisms and potential side effects of using SBE-β-CD as a drug delivery system for TQ.

## 1. Introduction

Recently, natural products, medicinal plants, their biologically active metabolites, and many food additives have been widely studied for their therapeutic applications such as anti-inflammation, immunomodulation, and anti-cancer effects. Among these, *Nigella sativa* (black seed) is a well-known medicinal plant that has been extensively researched in the pharmaceutical and nutraceutical fields [1]. Black seed has gained popularity in the Arab world, especially in Gulf countries, due to its natural and beneficial properties for health [2].

Thymoquinone (Figure 1), 2-Isopropyl-5-methyl-1, 4-benzoquinone, is the main chemical constituent of the black seed, with a chemical formula of C_10_H_12_O_2_ and molecular mass of 164.204 g/mol. The concentration of TQ in seed oil has been reported to be between 18 and 25 µg/mL [3]. TQ has demonstrated anti-proliferative effects for different cancer types [4] and is considered a promising anticancer candidate either alone or in combination with chemotherapeutic agents [5,6,7,8]. Additionally, TQ exhibited antioxidant, anti-inflammatory, and apoptotic activities in vitro and in vivo [9].

TQ belongs to monoterpenes with keto-enol tautomerism, and its keto form (Figure 1) has been reported to possess pharmacological activities. The solubility of TQ in aqueous solutions ranges from 549 to 669 mg/mL [10]; however, its stability is compromised by factors such as pH and light [11]. To overcome these challenges, a well-designed inclusion complex using cyclodextrin should be proposed to improve the solubility, stability, and other physicochemical properties of TQ, which is considered a promising small molecule candidate for anticancer therapy.

Sulfobutylether-β-cyclodextrin (SBE-β-CD) is a polyanionic cyclic oligosaccharide that is non-hygroscopic and non-crystalline. It is generally formed by replacing the primary or secondary hydrogen of the hydroxyl groups of CDs with sulfobutyl groups. SBE-β-CD has many exceptional properties, including relatively high solubility (>50 folds compared to β-CD) and improved stability and biocompatibility, making it a functional biopolymer for various drug delivery applications with thermodynamically stable inclusion complexes. SBE-β-CD has provided a smart solution to the commercial development of many therapeutics by overcoming the major hurdles of the poor solubility and instability of pharmacologically active candidates, such as TQ [11]. However, the estimation of CDs (including SBE-β-CD) in different biological media must be investigated via stringent pharmacokinetic [12] and toxicological studies of SBE-β-CD-enabled drug formulations [13].

While there is evidence to suggest that TQ has potential anticancer properties, the studies investigating its effects on cancer cells vary in their methodologies, concentrations used, and types of cancer cells tested [14,15,16]. For instance, a study demonstrated that TQ suppressed the proliferation of breast cancer cells by inducing apoptosis and cell cycle arrest in a dose-dependent manner. The study reported that TQ inhibited the growth of breast cancer cells with an IC_50_ value of approximately 10–20 μM [17]. Similarly, another study found that TQ inhibited growth and induced apoptosis in pancreatic cancer cells with an IC_50_ value of approximately 10 μM [18].

In addition to its direct effects on cancer cells, TQ has also demonstrated the ability to enhance the effectiveness of chemotherapy and radiation therapy in cancer treatment [19]. A study reported that TQ sensitized colon cancer cells relative to chemotherapeutic drug 5-fluorouracil (5-FU) by inhibiting the expression of a protein that is known to confer drug resistance. The study found that the combination of TQ and 5-FU significantly reduced the viability of colon cancer cells compared to either treatments alone [20]. Despite its potential as an anticancer agent, the clinical application of TQ as an anticancer agent has been limited by its poor solubility and bioavailability. TQ is a hydrophobic (water-insoluble) compound, which makes it difficult for the body to absorb and utilize it effectively. This can result in lower therapeutic efficacy and increased toxicity [14].

In order to improve solubility and bioavailability, various delivery systems have been developed, including the use of nanoparticles as a delivery system [21]. For instance, an in vitro study found that TQ-loaded nanoemulsions had a higher solubility and improved anticancer activity compared to free TQ [22]. Similarly, another study found that TQ-loaded nanoparticles had a higher cellular uptake and greater anticancer activity compared to free TQ [23]. Despite the potential of TQ as an anticancer agent, further research is needed to optimize its delivery and improve its clinical effectiveness [14]. The concentration of TQ required for anticancer activity can vary depending on the type of cancer cells and experimental conditions used. Thus, more studies are necessary to determine the optimal concentration and delivery method for clinical use.

## 2. Results

### 2.1. Phase Solubility Study

Higuchi and Connors developed a phase solubility diagram that uses X and Y coordinates to explain the inclusion complexes of poorly water-soluble candidates as quests with cyclodextrins (CD) and as a host. This diagram examines the solubility profiles of these candidates and their stabilities constants to determine the efficacy of the inclusion complex’s formation [24]. Figure 2 displays the phase solubility diagram of TQ with SBE-β-CD in water at five different temperatures (293–318 K°), showing that the water solubility of TQ increased linearly with the increase in the molar concentrations of SBE-β-CD. This indicates that this system follows an A_L-_type solubility diagram [25] with an independent degree of substitution [26]. The apparent stability constants (K1:1) of the inclusion complexes were calculated from the straight-line equation of the phase solubility curve for each separate temperature studied (293–318 K°), and they were found to be 1480–8535 M^−1^, indicating the formation of a favorable inclusion complex between TQ and SBE-β-CD. The solubility of TQ in water was observed to increase from 0.0154 mM to 0.4369 mM at 310 K (37 °C), indicating that the solubility of TQ in water was greatly improved by the successful formation of the inclusion complex between TQ and SBE-β-CD. The efficiency (CE) of the complexation, defined as the ratio of the concentration of the dissolved complex to the concentration of the dissolved free CD, was well-reported by Loftsson et al. and was used to evaluate the solubilization of hydrophobic molecules in cyclodextrin [27]. From the phase solubility diagram, the CE for the TQ-SBE-β-CD inclusion complex was found to be 1.030, indicating that SBE-β-CD is suitable for the formulation of TQ.

Figure 3 shows the Van’t Hoff plot for the TQ-SBE-β-CD complex, which exhibits a linear relationship between lnK_f_ V_s_ and the inverse of the absolute temperature (K). Table 1 presents the thermodynamic parameters, including enthalpy change (ΔH), the entropy change (ΔS), and the Gibbs free energy change (ΔG). The negative values of ΔH indicate that the interaction processes between TQ and SBE-β-CD are exothermic. The high value of ΔH can be explained by strong interactions, including van der Waals bonds and hydrophobic interactions, which are typically associated with high-energy systems. These interactions arise from the dehydration of guest molecules and the penetration of hydrophobic guests into the cyclodextrin cavity.

### 2.2. X-ray Diffraction

The inclusion complex of TQ with SBE-β-CD was investigated further by using XRD. Figure 4 shows the X-ray diffraction patterns of pure TQ, pure SBE-β-CD, a physical mixture, and their corresponding inclusion complexes. There were no diffraction peaks in the X-ray spectra of SBE-β-CD, while sharp diffraction peaks were obtained in the pure TQ, indicating the crystalline state of the molecule. The same results were observed in the physical mixture but with less intensity. However, the inclusion complex of TQ with SBE-β-CD exhibited well-differentiated diffused X-ray patterns, indicating the amorphous characteristics of the complex. The lack of the crystallinity of TQ is considered clear evidence of the formation of the inclusion complex [28]. This phenomenon indicates an interaction between TQ and SBE-β-CD, ultimately leading to the formation of a new solid phase [28,29]. The X-ray diffraction patterns of the TQ-SBE-β-CD system were characterized by large diffraction peaks, making it impossible to differentiate between pure TQ as a crystal and SBE-β-CD. These findings indicate that TQ no longer exists as a crystalline form, and its solid complexes with SBE-β-CD exist in the amorphous state (Figure 4d). The formation of an amorphous state is significant evidence that TQ was dispersed in a molecular state within SBE-β-CD [30,31].

### 2.3. Fourier Transform Infrared Spectroscopy (FT-IR)

Figure 5 displays the FT-IR spectra of the TQ-SBE-β-CD inclusion complex, pure TQ, pure SBE-β-CD, and a physical mixture of TQ in SBE-β-CD. The spectra were analyzed to investigate the interaction of TQ with SBE-β-CD based on the characteristics of the IR peaks of each component. Pure TQ showed peaks at 2965.928–2875.083 cm^−1^ (Figure 5A), which correspond to the stretching vibrations bands of isopropyl and CH_3_ groups. The band at 2922.530 cm^−1^ represents the spectrum of the C-H stretching of tertiary carbon in the isopropyl group, and the 2875.083 cm^−1^ band indicates the symmetric stretching of three methyl groups. The SBE-β-CD showed a clear peak at 3368.656, 2926, 1643.605, 1152.034, 1024.975, and 520.617 cm^−1^ (Figure 5B). However, all these FT-IR bands were shifted to lower frequencies in the spectra of the inclusion complex, indicating that TQ was encapsulated in the SBE-β-CD cavity, and the vibration of TQ was restricted. Furthermore, all the peak bands of TQ disappeared from the inclusion complex spectra, indicating that the breakdown of intermolecular hydrogen bonding within TQ established a weak force in the complex system (Figure 5D).

### 2.4. Antiproliferative Effects of the Formulation

#### 2.4.1. Effect of TQ and TQ/SBE-β-CDs on Colorectal Cancer Cell Lines

Figure 6 shows the cell viability results of colon cancer cells, specifically HCT116 and HT29 cells, treated with pure TQ, pure SBE-β-CD, and the TQ/SBE-β-CD inclusion complex for 48 and 72 h. Treatment with SBE-β-CD alone (red line) did not show any inhibitory effect on either cell line at both time points compared to the control (blue line). Treatment with TQ alone resulted in the inhibition of HCT116 cell growth at both 48 h (Figure 6A) and 72 h (Figure 6C), with a greater effect observed at 72 h. At the highest concentration of TQ (100 µg/mL), cell viability was reduced to 27.8%, and the IC_50_ was 4.7 ± 0.21 µg/mL. Treatment with the TQ/SBE-β-CD inclusion complex for 72 h slightly increased the inhibitory effect on HCT116 cells (Figure 6C), with cell viability reduced to 20.66% at 100 µg/mL, and IC_50_ decreased to 1.2 ± 0.16 µg/mL.

Similarly to HCT116 cells, treatment with SBE-β-CD alone on the HT29 cells did not show any inhibitory effect at both time points. However, treatment with pure TQ resulted in a dose-dependent inhibitory effect at both time points, with the strongest inhibition observed after 72 h, where the highest concentration (100 µg/mL) resulted in a cell viability of 16.58% (Figure 6D) and an IC_50_ of 0.8 ± 0.10 µg/mL. Combining TQ with SBE-β-CDs as an inclusion complex after 72 h resulted in an increase in the inhibitory effect compared to TQ alone, with cell viability reduced to 13.8% and IC_50_ decreased to 0.4 ± 0.02 µg/mL (Figure 6D). Overall, among the two colon cancer cell lines tested, a higher inhibitory effect of TQ/SBE-β-CD was observed in HT29 cells.

#### 2.4.2. Effect of TQ and TQ/SBE-β-CD on Breast Cancer Cell Lines

In this study, the response of breast cancer cell lines, including MDA-MB-231, SkBr3, and MCF7, to SBE-β-CD, pure TQ, and the TQ/SBE-β-CDs inclusion complex was investigated. MDA-MB-231 cells showed growth inhibition after treatment with pure TQ at both 48 h and 72 h (Figure 7). In this study, 100 µg/mL of TQ reduced the cell viability of MDA-MB-231 cells to 50.23% and 27.16% after 48 h and 72 h treatment, respectively. The IC_50_ of the TQ treatment in the MDA-MB-231 cell line at 72 h was detected as 3.3 ± 1.99 µg/mL. The TQ/SBE-β-CD treatment at 72 h enhanced the inhibitory effect by reducing the viability percentage of MDA-MB-231 cells to 24% and decreasing the IC_50_ to 0.9 ± 0.36 µg/mL compared to TQ. SkBr3 cells responded to TQ treatment for 48 h and 72 h by generating an IC_50_ of 1.3 ± 0.08 µg/mL and 0.2 ± 0.01 µg/mL, respectively. TQ/SBE-β-CD enhanced the inhibitory effect on SkBr3 cells by decreasing the IC_50_ to 0.70 ± 0.11 µg/mL at 48 h. MCF7 cells also responded to TQ and TQ/SBE-β-CD treatments with a similar pattern. The IC_50_ in response to the 48 h TQ treatment on this cell line was detected as 18.7 ± 1.27 µg/mL at 48 h and 0.7 ± 0.10 µg/mL at 72 h. The TQ/SBE-β-CD treatment after 48 h resulted in 5.1 ± 0.26 µg/mL and 0.40 ± 0.06 µg/mL after 72 h. The greatest inhibitory response to the TQ/SBE-β-CD treatment among the three breast cancer cell lines was detected in the SkBr3 cell line at both 48 h and 72 h, where the highest concentration (100 µg/mL) resulted in 23.1% and 6.4% cell viability after 48 h and 72 h treatment, respectively.

#### 2.4.3. Effect of TQ and TQ/SBE-β-CD on HepG2 Liver Cancer Cells

Figure 8 shows the percentage of cell viability against HepG2 liver cancer cells treated with SBE-β-CD, TQ, and TQ/SBE-β-CD at 48 and 72 h time points and different concentrations. SBE-β-CD had no significant inhibitory effects at either time point compared to the control, while TQ (green line) demonstrated a dose-dependent growth inhibitory effect at both time points, followed by even greater viability suppression in TQ/SBE-β-CD-treated cells (purple line). After 48 h of treatment, TQ at a concentration of 100 µg/mL resulted in 38.0% cell viability and an IC_50_ of 24.7 ± 5.85 µg/mL, whereas TQ/SBE-β-CD-treated cells had a significantly reduced viability percentage of 27.36% and an IC_50_ of 8.5 ± 0.99 µg/mL. Additionally, after 72 h of treatment with TQ/SBE-β-CD at a concentration of 100 µg/mL, the viability percentage was reduced to 11.5%, and IC_50_ decreased to 0.4 ± 0.03 µg/mL. The figure illustrates that at both time points, higher concentrations of TQ/SBE-β-CD resulted in lower cell viability percentages. Figure 9 represents the differences in the IC_50_ values obtained from the TQ and TQ/SBE-β-CD treatments across various cell lines. At 48 h, the IC_50_ of TQ/SBE-β-CD treated cells was lower than the TQ-treated cells in all cell lines. This trend was also observed at 72 h for HCT116 cells, and although the other cell lines also showed a lower IC_50_ in the TQ/SBE-β-CD treatment compared to TQ, the effect was not statistically significant at 72 h.

### 2.5. Molecular Modeling

To gain a deeper understanding of the mechanism of the inclusion of TQ into the cavity of SBE-β-CD, we performed theoretical calculations using quantum mechanics methods both in the gas phase and in the solution. Due to the large size of the complexes, we optimized the structure of the complexes using PM6-D3H4 in the solution and the gas phase, accounting for dispersion and hydrogen bonding. The optimized geometries for both models of the complexes are presented in Figure 10, which clearly shows that in both orientations, the guest molecule is fully embedded inside the cavity of cyclodextrin, indicating favorable interaction. The results of the binding energy are summarized in Table 2, showing substantial negative binding energy for both models of complexes, indicating that the complexation process is favorable. Gas phase calculations were further refined using DFT-ωB97X-D functionality with BSSE corrections, which produced highly stable complexes. The complexation energy in the aqueous media and the gas phase favors the formation of the complex by entering the methyl group inside the cavity of the cyclodextrin (complex A) by 4.1 kcal mol^−1^ and 16.5 kcal mol^−1^, respectively. Complexation is spontaneous and enthalpy driven. The hydrophobic guest’s affinity for the relatively hydrophobic cavity of SBE-β-CD leads to the formation of a stable complex. The location of one of the oxygen atoms of TQ near the wider portal of SBE-β-CD allows for the formation of strong hydrogen bonding, which increases stability. The negatively charged cyclodextrin is highly soluble in water; consequently, the formation of a highly stable complex with TQ enhances the solubility and bioavailability of this guest.

## 3. Discussion

Cancer is a major public health concern and is currently the second leading cause of human mortality worldwide after cardiovascular diseases. While anticancer drugs are widely used in cancer treatments, they are often associated with serious side effects. Therefore, it is crucial to develop new cancer treatments [32]. TQ is a natural bioactive compound known for its anticancer properties. These properties are attributed to its ability to intervene with different oncogenic pathways, suppress inflammation and oxidative stress, prevent angiogenesis and metastasis, and induce apoptosis. TQ has also been shown to induce cell cycle arrest at the G1 phase in human breast and colon cancer via upregulating p21 and p27 cyclin-dependent kinase inhibitors (CDK) and suppressing cyclin E and D. Moreover, TQ’s inhibitory effect on the cell cycle is dose-dependent, where a high dose of TQ induced the G2 phase in MCF7 breast cancer cells while a low dose induced the S phase arrest in this cell line [33].

Another study reported that TQ, a bioactive compound of Nigella sativa (black cumin), exhibited anticancer effect properties via the induction of apoptosis by upregulating Bax, caspase-3, 8, and 9 and downregulating anti-apoptotic Bcl-2. Moreover, TQ showed an inhibitory effect against Akt, which is known to contribute to cell proliferation. TQ was also found to inhibit the NF-kappa B (NF-κB) signaling pathway, thereby suppressing inflammation and angiogenesis, which subsequently suppressed cancer. In addition, TQ exhibited an antioxidant effect, thus protecting cells from cancer [34].

Despite its promising anticancer pharmacological effect, TQ has limited clinical access due to its low solubility and absorption. Therefore, a nanoformulation approach to enhance the solubility of TQ is important, as highlighted in a recent study [35]. One strategy to improve the aqueous solubility of poorly soluble drugs is the use of natural cyclodextrins (CD), which also enhances drug stability [36]. In our study, SBE-β-CDs were selected and tested in combination with TQ to assess the potential application of TQ as an anticancer agent. In addition, SBE-β-CDs were selected due to their high solubility and the Food and Drug Administration (FDA) approval, which facilitates future commercialization [32]. In our study, it was found that the water solubility of TQ increased linearly with the increase in the molar concentrations of SBE-β-CD, indicating the formation of a favorable inclusion complex. The solubility of TQ in water was greatly improved by the successful formation of the inclusion complex, and the efficiency of complexation was found to be high, indicating that SBE-β-CD is suitable for the formulation of TQ. Overall, these results suggest that the use of CD as hosts can be a promising approach for improving the solubility and bioavailability of poorly water-soluble compounds such as TQ.

The enthalpy of inclusion is primarily dependent on the stabilization of complexation and the release of water, which contribute to the solvation of the nonpolar guest molecule [37]. Higher interaction strength leads to the increased stabilization of guest–host complexes. In addition, the negative value of entropy change (ΔS) confirms that the inclusion of TQ inside the nanocavity of SBE-β-CD results in a decrease in the translational and rotational degrees of freedom with respect to the encapsulated molecules, leading to a more ordered system. These findings further support the conclusion that the inclusion of TQ with SBE-β-CD has occurred. The negative value of ΔG also indicates that the inclusion process is spontaneous.

The selection of SBE-β-CD as a host candidate over the other CD derivatives is due to the novelty in solubilization using CD, particularly the SBE-β-CD-based product instead of HP-β-CD [13,26,38], due to the many attractive physicochemical characteristics of SBE-β-CD. Das et al. reported very interesting facts regarding the average number of substituents that reacted with the CD molecule, specifically the number of hydroxyl groups (-OH) with SBE-β-CD derivatives having a DS of 6.2–6.9 [39]. The result of our study showed that the TQ/SBE-β-CD treatment induced a stronger inhibitory effect on all cancer cells compared to TQ-treated cells. Additionally, TQ/SBE-β-CD also resulted in a lower IC_50_ compared to TQ in all cells at both 48 h and 72 h. Among the breast cancer cells tested, the inhibitory effect of TQ/SBE-β-CD was more pronounced on SkBr3 cells. It is worth noting that SkBr3 cells are estrogen- and progesterone-independent and do not express estrogen and progesterone receptors (ER and PR). Moreover, SkBr3 cells highly express human epidermal growth factor receptor 2 (HER2), making them more sensitive to treatment compared to MCF7 cells, which are ER+, PR+, and HER2− [40]. In contrast, MDA-MB-231 is a highly invasive, aggressive, and poorly differentiated triple-negative breast cancer (TNBC) cell line due to its lack of expression of ER, PR, and HER2 receptors [41]. This lack of expressions may explain why MDA-MB-231 cells were less responsive to the TQ/SBE-β-CD treatment compared to other breast cancer cells in this study. Ruthenium(II) arene complexes have gained attention for their catalytic activity in oxidizing NADH to NAD+, which increases ROS production and ultimately induces apoptosis in cells. Among these complexes, Ru(p-cym)(2-ampy)PPh3(PF6)2, a member of this class of complexes, has demonstrated potent cytotoxic effects against MCF7 cells, with IC_50_ values as low as 3.41 ± 2.87. The incorporation of a triphenylphosphine ligand into the coordination sphere of these complexes may further enhance their anticancer properties. Therefore, investigating the potential synergy between the TQ/SBE-β-CD treatment and Ruthenium(II) arene complexes, such as Ru(p-cym)(2-ampy)PPh3(PF6)2, warrants further investigation for the development of more effective cancer therapies [42].

The response of HCT116 and HT29 colon cancer cells to the TQ/SBE-β-CD treatment was also evaluated in this study. After 48 h and 72 h of treatment, both cell lines showed a growth inhibitory effect; however, the IC_50_ was lower in HT29 compared to HCT116, suggesting that a lower concentration of TQ/SBE-β-CD was required to inhibit 50% of HT29’s cell growth. Although both cell lines are colon cancer cells, HT29 exhibited a stronger response to the TQ/SBE-β-CDs treatment than HCT116. The differential sensitivity of these two cell lines to a particular treatment could be attributed to various factors, including differences in their molecular characteristics, signaling pathways, and cellular microenvironment. Both HT29 and HCT116 are colon cancer cell lines, but they have distinct genetic and molecular features that may affect their sensitivity to different treatments. HCT116 is a wild type for the tumor suppressor gene (p53) with other mutations in the K-RAS and PI3KCA genes [43], while HT29 had a mutation in p53 [44]. These differences in genetic and molecular features can influence the response of the cells to a specific treatment. Moreover, the level of oxidative stress and metabolic requirements may also play a role in the differential sensitivity of HT29 and HCT116 cells to TQ/SBE-β-CD. TQ has been shown to induce oxidative stress in cancer cells, which can lead to cell death [45]. Therefore, HT29 cells may be more susceptible to oxidative stress-induced cell death due to their higher basal levels of reactive oxygen species, such as H_2_O_2_ (hydrogen peroxide) emission, compared to HCT116 or normal cells, leading to a lower glutathione-buffering capacity and the activation of caspase 3, a protein involved in apoptosis [46]. Thus, HT29 maybe is more susceptible to oxidative stress-induced cell death compared to HCT116 cells, and this could explain why HT29 cells exhibit greater sensitivity to the TQ/SBE-β-CD treatment.

Treatment with TQ/SBE-β-CD significantly reduced the IC_50_ dose after 48 h compared to TQ against HepG2 liver cancer cells. The sensitivity of HepG2 cells to the TQ treatment may be due to several factors. One possible explanation is that TQ induces apoptosis in HepG2 cells [47]. In addition, TQ exhibits other anti-cancer properties, such as inhibiting cell proliferation, migration, and invasion [48], as well as inducing cell cycle arrest [49] and suppressing tumor angiogenesis [50]. The sensitivity of HepG2 liver cancer cells to TQ may be influenced by several mechanisms. For instance, SBE-β-CD may enhance the solubility of TQ and increase its cellular uptake, which allows for better targeting of cancer cells [51]. Since TQ has limited solubility in water, its effectiveness can be hampered [52]; however, SBE-β-CD can help overcome this limitation [53].

Our theoretical calculations using quantum mechanics methods support the experimental evidence of TQ’s inclusion in SBE-β-CD’s cavity, providing a deeper understanding of the mechanism of inclusion and the favorable interaction between the guest molecule and cyclodextrin. Overall, this study suggests that the use of SBE-β-CD may improve the anticancer effect of TQ on cancer cells by increasing its solubility and bioavailability and cellular uptake. Nonetheless, it is crucial to emphasize that further studies are needed to comprehend the underlying mechanisms and potential adverse effects of using SBE-β-CD as a drug delivery system for TQ. These studies could include investigating the long-term effects of using SBE-β-CD as a drug delivery system, determining the optimal dosage of TQ that can be delivered using SBE-β-CD, evaluating the long-term stability of the inclusion complex, and studying the release kinetics and release mechanism of TQ from the inclusion complex in different media. Additionally, it may be valuable to assess the pharmacokinetic, pharmacodynamic behavior and the anti-tumorigenic effect of the inclusion complex in vivo and evaluate any potential toxicity associated with its use.

## 4. Materials and Methods

### 4.1. Materials

Thymoquinone (CAS No. 490-91-5., Cat. No.: HY-D0803) with a purity of 99.59% and sulfobutylether-β-cyclodextrin (CAS No. 182410-00-0) were both purchased from MedChem Express (Monmouth Junction, NJ, USA). Acetonitrile, methanol, potassium dihydrogen orthophosphate, and orthophosphoric acid were purchased from Sigma-Aldrich (Taufkirchen, Germany). All solvents, including purified water, were of high purity and analytical grade.

### 4.2. Phase Solubility Study

Phase solubility studies were conducted following the method described by Higuchi and Connors [24]. An excess amount of TQ (10–15 mg) was mixed in a series of aqueous solutions containing increasing concentrations of SBE-β-CD (0–0.01 M) and dispersed using a vortex mixer for 3 min. The mixtures were then oscillated using a rotary shaker (MRC, Essex, UK) at 20–45° for 72 h to achieve equilibrium. After that, the samples were filtered through a 0.45 μm nylon filter (waters, Milford, MA, USA) and appropriately diluted. Then, the concentration of the dissolved TQ was determined by HPLC (Dionex, Thermo system, Waltham, MA, USA). The presence of SBE-β-CD did not interfere with HPLC measurements. The apparent stability constants (K) and complexation efficiency (CE) were calculated from the phase solubility diagram using Equations (1) and (2), respectively:(1)$$K=\frac{Slope}{S_{0}.\;\left(1-Slope\right)}$$
(2)$$\mathrm{CE}=\frac{Slope}{\left(1-Slope\right)}$$
where *S*_0_ represents the solubility of TQ in the absence of SBE-β-CD, and slope refers to the corresponding slope of the phase solubility diagrams.

### 4.3. Preparation of Inclusion Complex

To prepare the inclusion complex, a freeze-drying method was employed. Thymoquinone (0.2183 g) and SBE-β-CD (2.146 g) were dissolved in 20 mL of ultra-pure water at a 1:1 molar ratio. The mixture was mixed in a rotary shaker at room temperature for 72 h and then filtered through a 0.45 μm filter. The clear solution was frozen at −80 °C and subjected to freeze drying at −55 °C for 24 h.

### 4.4. Preparation of Physical Mixture

To prepare a physical mixture of TQ and SBE-β-CD in the same weight ratio as the lyophilized complex, TQ and SBE-β-CD were combined using a mortar and pestle for 7 min until a homogenous powder was obtained.

### 4.5. High-Performance Liquid Chromatography

The chromatographic system consists of Dionex Ultimate 3000 (Thermo Scientific, Waltham, MA, USA) equipped with a photodiode array detector set at a wavelength of 254 nm. The separation was performed on a hypercil gold column (4.6 mm I.D. × 250 mm length, 5 µm particle size; (Thermo Scientific, USA)) and with the column oven, which was maintained at 20 °C. Chromatographic separation was achieved successfully using a mobile phase composed of acetonitrile: 0.01 M potassium dihydrogen orthophosphate (65:35), with a pH of 4.5, adjusted by orthophosphoric acid. The mobile phase was delivered at a flow rate of 1 mL/min.

### 4.6. X-ray Diffraction (XRD)

The samples in powder forms were packed into the X-ray holder from the top before analysis. X-ray powder diffraction patterns were recorded using an X-ray diffractometer system (Rigaku, Austin, TX, USA), utilizing Cu-kα (λ = 1.5406 A) radiation, with a voltage of 40 kV and 40 mA current. The continuous scanning mode was applied at the rate of 2°/min over the range of 20–80°.

### 4.7. Fourier Transform Infrared Spectroscopy (FT-IR)

The FT-IR spectra were performed using Agilent resolution Pro (Santa Clara, CA, USA) with a potassium bromide (KBr) disc pan. The measurements were conducted within the scanning range of 4000–400 cm^−1^ at ambient temperature. The comparative FT-IR spectral data of the complexes, pure TQ, SBEβCD, and prepared physical mixture were discussed.

### 4.8. Cell Culture

Human colorectal carcinoma cells (HCT-116 and HT-29), human breast cancer cells (MDA-MB-231, SK-BR-3, and MCF-7), and human hepatoma cancer cells (HepG2) were cultured in complete DMEM supplemented with 10% FBS and 1% penicillin-xtreptomycin. The cells were cultured in T75 flasks and incubated in a humidified incubator at 37 °C with 5% CO_2_.

### 4.9. Cell Viability Assay

All cell lines cells seeded into a 96-well plate at a density of 5000 cells/well. After incubation with the coating solution overnight in a humidified atmosphere at 37 °C with 5% CO_2_, the media were removed. Then, the cells were treated with TQ or TQ/SBE-β-CD at a range of concentrations (0.0013, 0.0064, 0.032, 0.16, 0.8, 4, 20, and 100 µg/mL) at two-time points 48 h and 72 h. The untreated cells were used as a control. After that, 10 μL of 5 mg/mL MTT solution was added to each well [54]. The cell viability of SBE-β-CD was tested at concentrations similar to the complex. At the end target time point, the cells were incubated for 4 h before adding 100 μL of DMSO into each well. The absorbance reading of the plate was read at a wavelength of 570 nm using a microplate reader.

### 4.10. Molecular Modeling

Figure 1 shows the chemical structure of TQ, for which its initial geometry was optimized using the DFT-B3LYP method with a 6-31G(d,p) basis set. The structures of β-cyclodextrin (βCD) were extracted from the crystallographic parameters provided by the Structural Data Base System of the Cambridge Crystallographic Data Center and were optimized by minimizing their energy using the PM6-D3H4 semiempirical method with MOPAC 2012 software [55,56]. The PM6-D3H4 method implements corrections to hydrogen bonding and dispersion. The structure of SBE-β-CD was constructed using the β-CD skeleton by substituting six sulfobutylether moieties randomly at the primary OH groups of the narrow rim on the glucose units of the β-CD. The geometry of the SBE-β-CD was further optimized by the PM6-D3H4 semiempirical method in both the gas phase and aqueous media.

In this work, we considered two geometries for the 1:1 inclusion complex of TQ with SBE-β-CD. The first geometry (complex A) was obtained by inserting TQ into the CD cavity with the propyl group facing the wide rim. The second geometry (complex B) was obtained with the methyl group facing the wide rim. The structures of the complexes of TQ and SBE-β-CD were further optimized by the PM6-D3H4 semiempirical method, and frequency analysis at the same level of theory confirmed that the stationary point corresponds to a minimum due to the absence of imaginary frequencies. To further investigate the stability of these complexes, we ran DFT calculations on the optimized complexes using the ωB97X-D/6–31G(d) level of theory [57]. Hybrid functional methods including empirical dispersion with basis set superposition error (BSSE) corrections were computed using the Boys counterpoise method at the same level [58,59]. The thermodynamic parameters such as enthalpy change (ΔH), entropy change (ΔS), Gibbs free energy change (ΔG), and binding energy of complexes (ΔE) were obtained from the difference between the values of the resulting complexes and those of the isolated host and guest.

### 4.11. Statistical Analysis

Statistical analysis was performed using GraphPad Prism (v. 9.5) to compare the IC_50_ values for different treatments at two time points. A one-way ANOVA test with Tukey post hoc was conducted, with a significance level of *p* < 0.05. Any *p* values below this threshold were considered to indicate a statistically significant difference between the treatments.

## 5. Conclusions

Cancer is a major public health concern worldwide, and the development of new treatments is critical. TQ is a natural bioactive compound that has been shown to possess anticancer properties. TQ exerts its effects by modulating oncogenic pathways, suppressing inflammation and oxidative stress, preventing angiogenesis and metastasis, and inducing apoptosis. However, limited clinical access to TQ is due to its low solubility and absorption. Therefore, a nanoformulation approach, such as using SBE-β-CD, was shown to be more effective than TQ alone in inhibiting cancer cell growth. This study has demonstrated that TQ/SBE-β-CD is more effective than TQ alone in inhibiting the cell growth of breast and colon cancer cell lines. The inhibitory effect of TQ/SBE-β-CD was the strongest on SkBr3 and HT29 cell lines, which are estrogen- and progesterone-independent and have distinct genetic and molecular features. Therefore, TQ/SBE-β-CD has the potential to be a promising strategy for cancer treatment, and further studies are warranted to explore its potential clinical applications. More research is needed to determine the optimal dose of TQ/SBE-β-CD for cancer treatment, and it may be necessary to conduct pharmacokinetic studies to evaluate the in vivo absorption and distribution of TQ/SBE-β-CD.

## Figures and Tables

**Figure 1 molecules-28-04096-f001:**
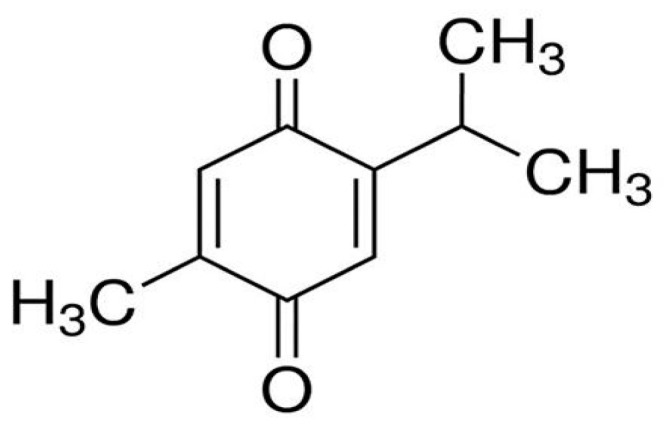
Thymoquinone structure.

**Figure 2 molecules-28-04096-f002:**
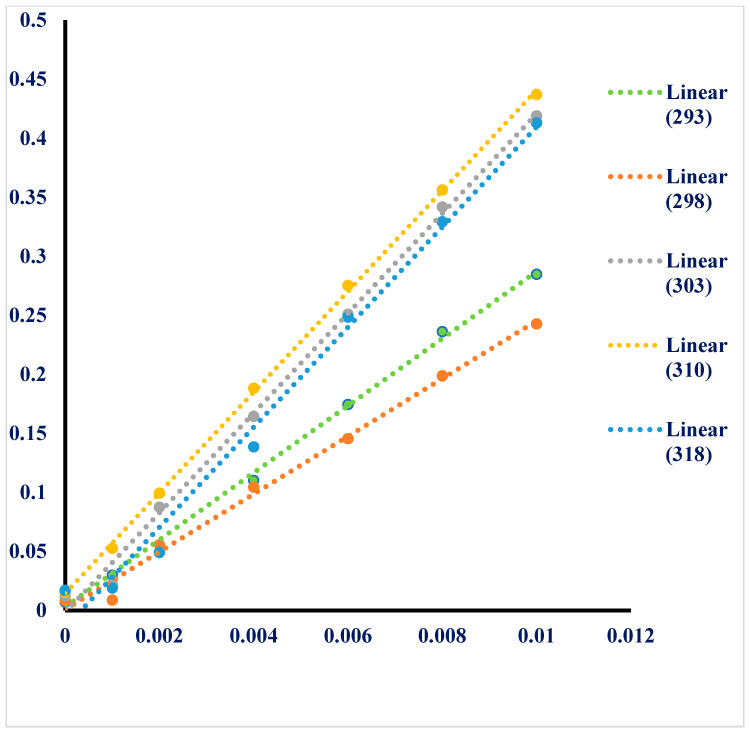
Phase solubility diagram for SBE-β-CD- TQ host–quest system at five different temperatures: 293 K, 298 K, 303 K, 310 K, and 318 K.

**Figure 3 molecules-28-04096-f003:**
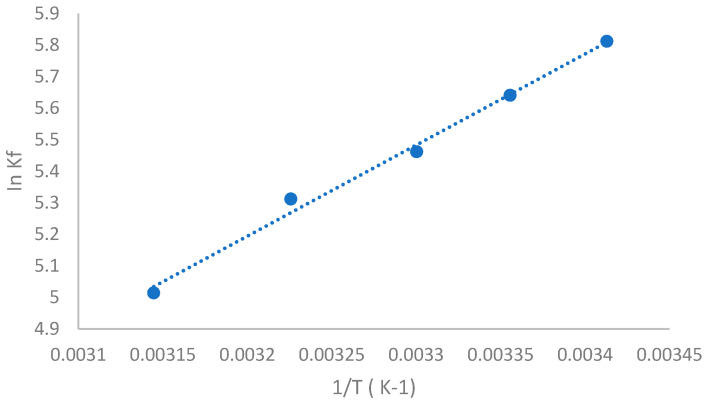
The relationship between InK_f_ and the inverse of the absolute temperature (1/T) for TQ-SBE-β-CD.

**Figure 4 molecules-28-04096-f004:**
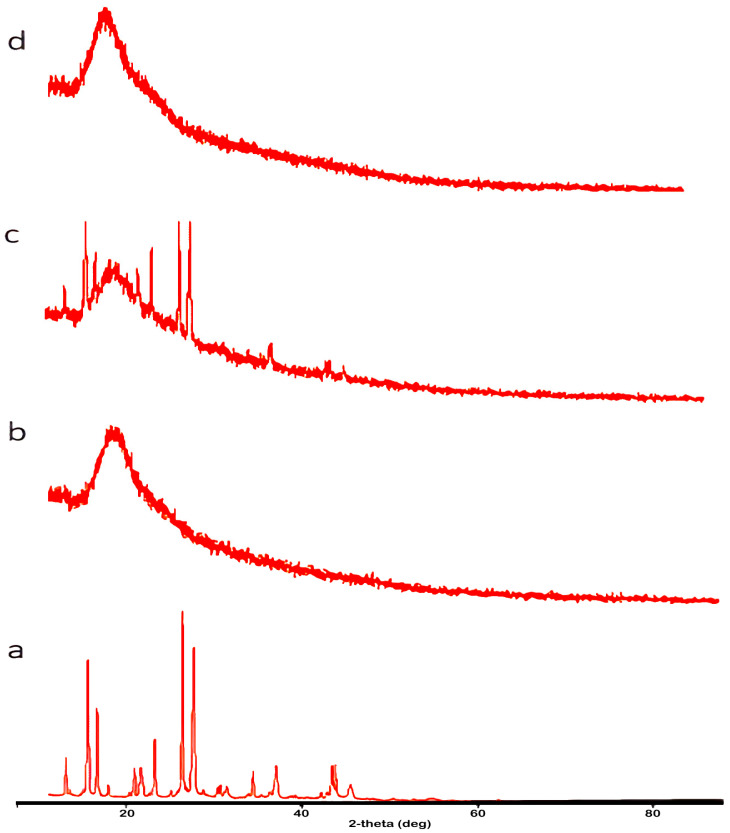
Powder X-ray diffractogram: (**a**) pure TQ. (**b**) Pure SBE-β-CD. (**c**) Physical mixture. (**d**) Inclusion complex.

**Figure 5 molecules-28-04096-f005:**
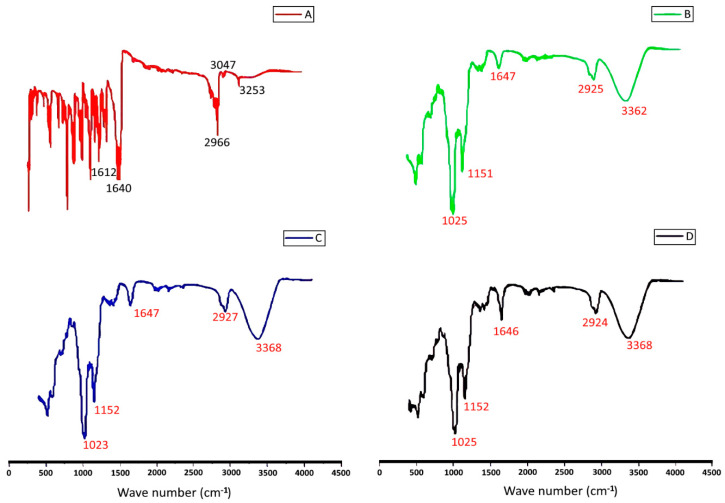
FT-IR spectra of (**A**) pure TQ. (**B**) Pure SBE-β-CD. (**C**) Physical mixture. (**D**) Inclusion complex.

**Figure 6 molecules-28-04096-f006:**
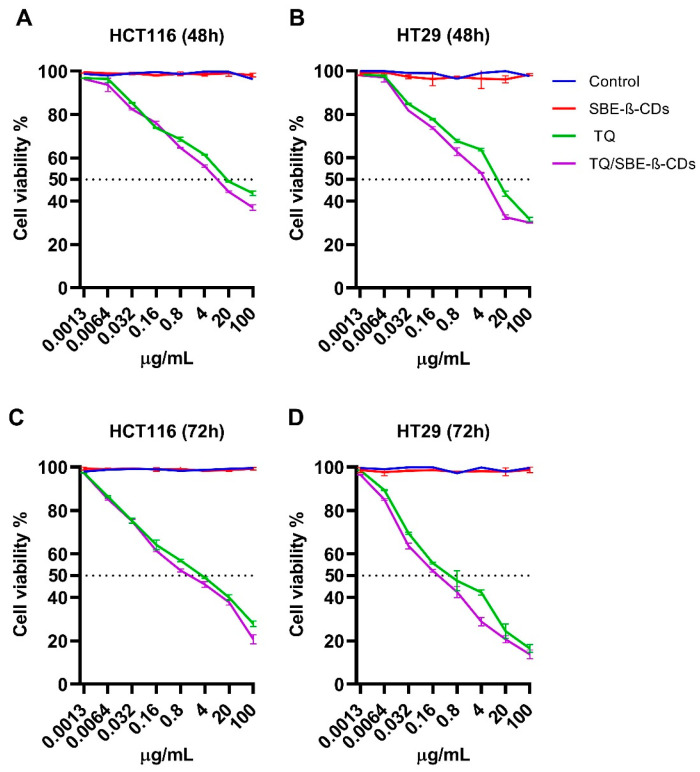
Cell viability of HCT116 and HT29 colon cancer cells treated with SBE-β-CD, TQ, and the TQ/SBE-β-CD inclusion complex. (**A**,**B**) represent the cell viability of HCT116 and HT29 at 48 h and (**C**,**D**) at 72 h, respectively. Data were expressed as the mean ± standard deviation (SD). The control group was left untreated.

**Figure 7 molecules-28-04096-f007:**
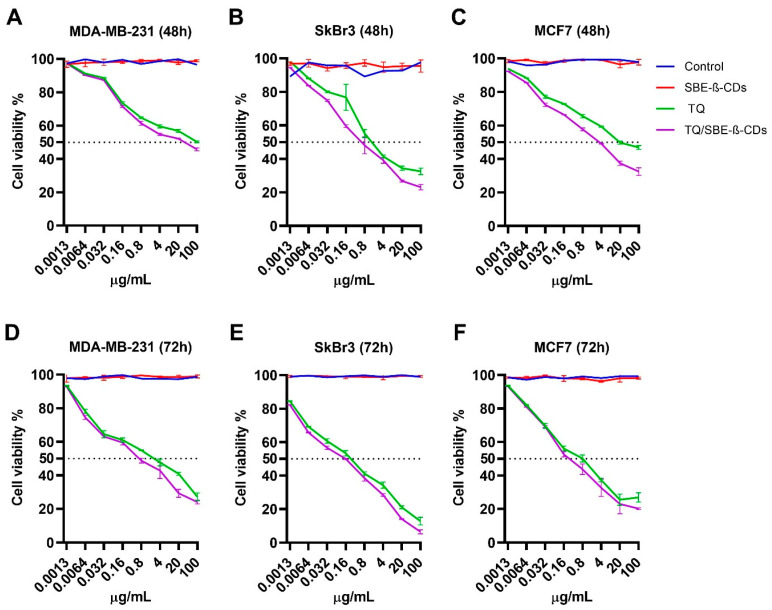
Cell viability of MDA-MB-231, SkBr3, and MCF7 breast cancer cells treated with the SBE-β-CD, TQ, and TQ/SBE-β-CD inclusion complex. (**A**–**C**) represents cell viability at 48 h, and (**D**–**F**) represents cell viability at 72 h. Data were expressed as the mean ± standard deviation (SD). The control group was left untreated.

**Figure 8 molecules-28-04096-f008:**
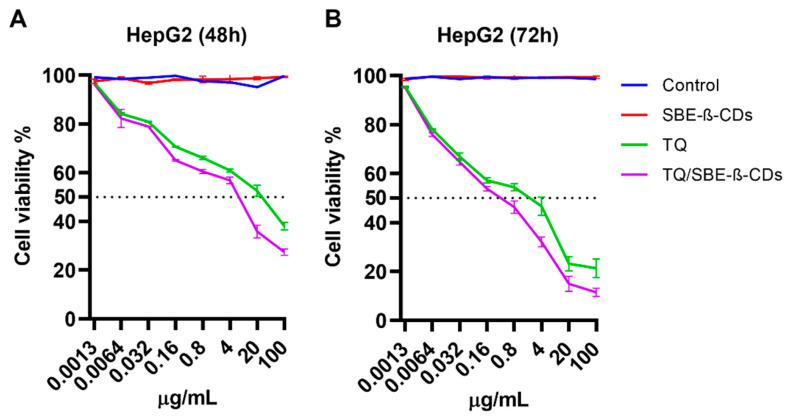
Cell viability % after treating HepG2 with different concentrations of SBE-β-CD, TQ, and the TQ/SBE-β-CD inclusion complex. (**A**,**B**) represent cell viability at 48 h and 72 h, respectively. Data were expressed as mean ± standard deviation (SD). The control group was left untreated.

**Figure 9 molecules-28-04096-f009:**
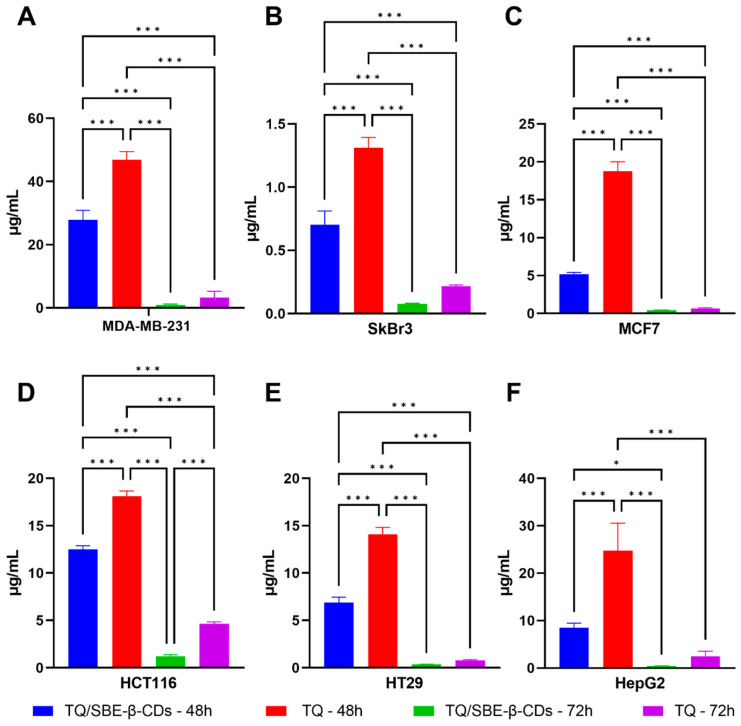
IC_50_ comparison between pure TQ and TQ/SBE-β-CD inclusion complex treatments on MDA-MB-231 cells (**A**), SkBr3 cells (**B**), MCF7 cells (**C**), HCT116 cells (**D**), HT29 cells (**E**), and HepG2 cells (**F**). Data are expressed as mean ± standard deviation (SD), * *p* < 0.05, *** *p* < 0.0001.

**Figure 10 molecules-28-04096-f010:**
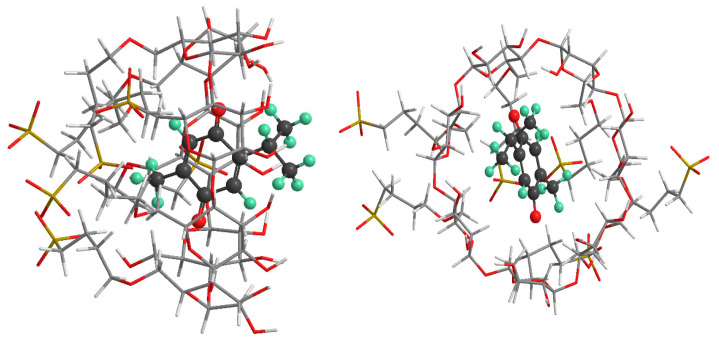
The optimized geometries of the complexes. The upper panel is for complex A, and the lower panel is for the complex B model of TQ with SBE-β-CD.

**Table 1 molecules-28-04096-t001:** The effect of temperature on thermodynamic parameters (K_f,_ ΔH, ΔS, and ΔG) of the inclusion complex of TQ-SBE-β-CD.

T (K)	Kf (L mol^−1^)	InKf	ΔS (J mol^−1^ K^−1^)	ΔH (kJ mol^−1^)	ΔG (kJ mol^−1^)
293	334.352	5.812	−33.7	−24.031	−9.850
298	1158.53	5.641	−33.7	−24.031	−10.018
303	640.162	5.462	−33.7	−24.031	−10.187
310	70.1386	5.312	−33.7	−24.031	−10.422
318	72.1245	5.014	−33.7	−24.031	−10.692

T: Temperature; ΔH: enthalpy change; ΔS: entropy change; ΔG: Gibbs free energy change.

**Table 2 molecules-28-04096-t002:** Thermodynamic parameters associated with the inclusion of TQ with SBE-β-CD in aqueous media using the PM6-D3H4 optimization method and the binding energy (ΔE) of these complexes in the gas phase calculated by the DFT- ωB97X-D function.

Compound	ΔH (kcal mol^−1^) *	ΔS(cal mol^−1^) *	ΔG(kcal mol^−1^) *	ΔE (kcal mol^−1^) **
Complex A	−544.1	−60.1	−526.1	−231.8
Complex B	−537.9	−53.4	−522.0	−215.3

* Calculations in the aqueous media using PM6-D3H4. ** ΔE for the gas phase calculation of the complexes using the DFT-ωB97X-D functionality with BSSE corrections. ΔH: Enthalpy change; ΔS: entropy change; ΔG: Gibbs free energy change; ΔE: binding energy.

## Data Availability

Not applicable.

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
