# Peer review of "Characterization of Thymoquinone-Sulfobutylether-β-Cyclodextrin Inclusion Complex for Anticancer Applications"

_molecules, 2023, doi:10.3390/molecules28104096_

Round 1

Reviewer 1 Report

The research article entitled “Characterization of Thymoquinone- Sulfobutylether-β-cyclodextrin Inclusion Complex for Anticancer Applications” was described by the Eid and his co-workers. In this paper, the authors have written the manuscript nicely, characterized and tested the new compound (TQ inclusion with a SBE-β-CD) for the treatment of cancer applications. There are a few comments are given at the bottom of this text.  

Comments:

1] Is Sulfobutylether-β-Cyclodextrins (SBE-β-CD) soluble in water at room temperature? If not, which temperature can it be dissolved in water?

2] Does the inclusion complex (TQ inclusion with a SBE-β-CD) have any NMR spectra?

3] There is no chemical structure of the inclusion complex given in the manuscript.

4] Page-14, line 406, “H2O2” should be written as “H2O2”.

Author Response

Thank you so much for your valuable comments ,the response is attached.  

Reviewer 2 Report

Recommendation: minor revision

Comments:

In the manuscript entitled "Characterization of Thymoquinone- Sulfobutylether-β-cyclodextrin Inclusion Complex for Anticancer Applications", Eltayeb E.M. Eid and Mohammed Abdullah Alshawsh et. al., reported antiproliferative activity of TQ after the inclusion complexes of thymoquinone with Sulfobutylether-β-cyclodextrin (SBE-ß-CD). The authors also explored the anticancer activity of TQ alone and complexed with SBE-ß-CD on six different cancer cell including colon, breast, and liver cancer cells (HCT-116, HT-29, MDA-MB-231, MCF-7, SK- BR-3, and HepG2), using MTT assay. The experiments are thoroughly conducted and the manuscript is well written. Though, there are some minor issues that the authors need to be considered while submitting the revised version. Thus, I would recommend this manuscript for publication in this esteemed journal after following modifications.

1. Can the authors explain why they have performed the experiments under four different temperatures (293-318 K)?

2. Can the authors performed the stability test for SBE-ß-CD under different biological medium?

3. “However, further studies are necessary to fully understand the underlying mechanisms and potential side effects of using SBE-ß-CD as a drug delivery system for TQ”-Can the authors explain what other sets of experiments are needed? And why the authors didn’t perform those experiments?

4. “Figure 2: Phase Solubility diagram for SBE-β-CD- TQ Host -Quest system at five different temperatures: 293 K, 298 K, 303 K, 310 K, 318 K”-can the authors change the color 293 K, and 318 K as they are confusing to the readers.

5. Table 1: why there is no change in ΔH with changing the temperature.

6. Can the authors provide a better resolution for Figure 4.

7. The authors should cite some of the recent relevant papers to compare their anticancer properties or/and IC50 values: ACS Appl. Bio Mater. 2022, 5, 1, 190–204; Dalton Trans., 2022, 51, 3937-3953; https://doi.org/10.1016/j.poly.2021.115379.

Can be improved.

Author Response

(The authors gave the same response as above.)

Reviewer 3 Report

The manuscript titled "Characterization of Thymoquinone- Sulfobutylether-β-cyclodextrin Inclusion Complex for Anticancer Applications" is very well written, organized and structured. Both Thymoquinone and Sulfobutylether-β-cyclodextrin have been extensively studied in the literature. However, the combination of both is what gives relevance to this study. Therefore, I suggest the acceptance of the document as long as some minor corrections are resolved.

·         In the third paragraph of page 2 reference 10 is repeated. Is it correct?

·         In section 2.1 check the acronyms K and °C.

·         In the manuscript they do not always use the acronym CD to refer to cyclodextrins, please homogenize it.

·         Line 167, section 2.2. Indicate when SBE-β-CD is pure.

·         In several parts of the manuscript, the authors define the acronyms

·         thymoquinone (TQ) and sulfobutylether-β-cyclodextrins (SBE-β-CD) should only be mentioned the first time they are mentioned in the text.

·         In FT-IR spectroscopy the authors should identify and assign the stretching bands of the C=O, C=C, =C-H and -C-H bonds in TQ pure. In the same sense, the authors must identify and assign the stretching bands of C-O, S=O, O-H and C-H in SBE-β-CD pure. Later, identify them in the physical mixture and in the inclusion complex.

·         In figure 5, no significant change is observed in the spectra of B, C and D, explain it in the text. Include a table with the values in wave number (cm-1) assigned to the main bonds or functional groups to see the changes.

·         In the acronym mL, the letter L should be capitalized.

·         In the acronym IC50, the number 50 must be as a subscript.

·         Line 208, section 2.4.1. Indicate when SBE-β-CD is pure.

·         The title of the X axis in figures 4 and 5 is unclear, improve the size and type of font.

·         Specify in sections 2.4.1 – 2.4.3, who is the drug used as control.

·         Why they did not evaluate the anticancer activity of the mixture of compounds. Explain it in the text. The results would have been very relevant.

·         Lines 268-272. The legend of figure 9 is misplaced.

Check in detail paragraph by paragraph, sentence by sentence the correct use of the English language.

Author Response

(The authors gave the same response as above.)
